# Effect of Mixed Starters on Proteolysis and Formation of Biogenic Amines in Dry Fermented Mutton Sausages

**DOI:** 10.3390/foods10122939

**Published:** 2021-11-29

**Authors:** Debao Wang, Guanhua Hu, Huiting Wang, Limei Wang, Yuanyuan Zhang, Yufu Zou, Lihua Zhao, Fang Liu, Ye Jin

**Affiliations:** 1Department of Food Science, College of Food Science and Engineering, Inner Mongolia Agricultural University, Hohhot 010018, China; Debaowang_2021@163.com (D.W.); 18247109171@163.com (G.H.); wht15847848708@163.com (H.W.); zlh15374887585@163.com (L.Z.); 2Institute of Agricultural and Livestock Products Processing, Inner Mongolia Academy of Agricultural and Animal Husbandry Sciences, Hohhot 010031, China; w524041348@163.com (L.W.); yy15598197155@163.com (Y.Z.); 3Zhengxiang White Banner Mengsheng Meat Industry Co., Ltd., Xilingol 013800, China; ms7ns8@163.com; 4Institute of Agricultural Products Processing, Jiangsu Academy of Agricultural Sciences, Nanjing 210014, China

**Keywords:** dry fermented sausage, starter cultures, mutton, proteolysis, biogenic amines

## Abstract

In this study, by comparing the four groups of sausages, namely, CO (without starter culture), LB (with *Lactobacillus sakei*), LS (with *L. sakei* 3X-2B + *Staphylococcus xylosus* SZ-8), and LSS (with *L. sakei* 3X-2B + *S. xylosus* SZ-8 + *S. carnosus* SZ-2), the effects of mixed starter cultures on physical–chemical quality, proteolysis, and biogenic amines (BAs) during fermentation and ripening were investigated. Inoculation of the mixed starter cultures increased the number of lactic acid bacteria and staphylococci in sausages during fermentation and ripening for 0 to 5 days. The *L. sakei* 3X-2B + *S. xylosus* SZ-8 + *S. carnosus* SZ-2 mixed starter accelerated the rate of acid production and water activity reduction of sausages and improved the redness value. Compared with CO, the mixed starter effectively inhibited *Enterobacteriaceae*. At the end of ripening, the LSS group was approximately 1.25 CFU/g, which was less than the CO group, thereby reducing the total volatile basic nitrogen (TVB-N) in the LSS group. The free amino acids in the LS and LSS groups (224.97 and 235.53 mg/kg dry sausage, respectively) were significantly (*p* < 0.001) higher than that in the CO group (170.93 mg/kg dry sausage). The level of histamine, cadaverine, putrescine, and common BAs showed an opposite trend to the increase of the corresponding precursor amino acid content, which were significantly lower (*p* < 0.001) in the LS and LSS sausages than in CO. This study showed that L. sakei 3X-2B + S. xylosus SZ-8 + S. carnosus SZ-2 is a potential mixed starter for fermented meat products.

## 1. Introduction

Fermented sausage is a popular and upscale processed meat product among consumers because of its unique color, texture, and flavor [1]. These typical characteristics are formed due to proteolytic, lipidolytic, and microbiological changes during fermentation and ripening of fermented sausage [2]. In the process, proteins are metabolized into peptides by cathepsin, and oligopeptides are metabolized into amino acids by aminopeptidase [3]. Some amino acids as precursor substances are metabolized into biogenic amines (BAs) by amino acid decarboxylase secreted by spoilage bacteria, such as *Pseudomonas* and *Enterobacteriaceae* [4,5]. BAs are composed of low-molecular-weight organic nitrogenous compounds with aliphatic, aromatic, or heterocyclic structures [6]. A moderate amount of BAs can promote normal physiological activities of the human body, but excessive intake of BAs may cause serious damage [7,8]. Based on the estimated safe levels of acute exposure to histamine (HIS) and tyramine (TYR) derived by the European Food Safety Authority [9], Torović et al. [10] calculated the exposure to BAs: 50 mg of HIS for healthy individuals, but below detectable limits for those with HIS intolerance; 600 mg of TYR for healthy individuals not taking MAOI drugs, but 50 mg/6 mg for those taking third-generation/classical MAOI drugs. A study conducted by Latorre-Moratalla et al. [11] concerning six European countries showed that eating 80 g of sausage could reach the lowest dangerous dose of TYR; HIS dose only had adverse effects on Dao patients. For the Austrian population, the maximum intake of BAs in fermented meat products was evaluated: adult female/male putrescine (PUT) 1.1/1.9 mg/day and cadaverine (CAD) 1.8/3 mg/day [12]. The TRPs for women and men were 23.4, 42.0, and 71.7 mg/day [13]. Therefore, restraining the content of various biogenic amines in fermented meat products has become the focus of consumers, especially patients.

BAs are known as potential precursor of carcinogenic nitrosamine substances, among which PUT and CAD could form heterocyclic nitrosamines with nitrite in acidic environments [14]. Changes of BAs in food are given high public concern. With regard to toxicity, HIS is the most important BA, which is related to the absorption concentration and presence of other amines that can enhance its toxicity, such as CAD, PUT, and TYR [15,16]. EFSA has recommended that the daily maximum intake of HIS and TYR is 50 and 600 mg/kg for healthy individuals, respectively [17]. Ruiz-Capillas et al. [18] reported that the consumption of foods containing high concentrations of biogenic amines, especially HIS and TYR, has been associated with health hazards, and effective measures should be taken to control the concentration of BAs to ensure high levels of food quality and safety.

Given the high BAs during processing of fermented food, the following measures are considered: (1) reducing the level of the BA precursors [19], (2) inhibiting the reproduction of amino acid decarboxylase-positive strains such as *Enterobacteriaceae* by adding antibacterial substances or antagonistic bacteria [11], and (3) using probiotics negative for amino acid decarboxylase as starter cultures [20,21]. Many scholars have studied the inhibition of BAs in products by inoculating starter cultures, which indicates the important role of BA oxidase [22,23]. Amine oxidase was synthesized by bioamine-degrading strains, which could decompose BAs into corresponding aldehydes, ammonia gas, and hydrogen peroxide [24,25]. Xie et al. [26] found that the amount of tryptophan (100%), phenylethylamine (PHE, 100%), PUT (86%), CAD (63%), HIS (82%), and TYR (43%) were significantly reduced by the mixed starter of *Lactobacillus plantarum* and *Staphylococcus xylosus*. Martuscelli et al. [27] studied the ability of *Staphylococcus xylosus*, isolated from artisanal fermented sausages in Southern Italy (Lucania region), to degrade BAs and found that S81 could degrade HIS, and its HIS oxidase activity was the highest, followed by S206 (93%), S79 (68%), and S90 (53%). Zhang et al. [28] studied the effect of *Lactobacillus plantarum* + *Lactobacillus salivarius* on the accumulation of BAs in traditional smoked horse meat intestines and found that they could catalyze oxidative deamination and promote the degradation of BAs by secreting amine oxidase. However, studies on the change of BAs during fermentation of mutton sausages were limited.

Wang et al. [29] used three standard strains of *L. sakei*, *S. xylosus*, and *S. carnosus* to explore their effects on the physical and chemical quality, proteolysis, and lipidolysis of fermented mutton sausages, which showed that the mixed starter of these three strains promoted the rapid acidification of sausages and increased free amino acid and fatty acid content. However, this study showed that the mixed starter promoted proteolysis while the total content of free amino acid was lower, which illustrated that short peptides and small molecules substances produced by protein decomposition during a 12 days processing cycle might turn into putrefactive ammonia substances and harmful biological amines. In view of the problems in the above research, we selected three starters from acidic gruel, fermented meat products, and air-dried meat through preliminary screening, re-screening, and 16S RNA identification, and named them *L. sakei* 3X-2B, *S. xylosus* SZ-8, and *S. carnosus* SZ-2. Previous studies have shown that the three strains *L. sakei* 3X-2B, *S. xylosus* SZ-8, and *S. carnosus* SZ-2 have good biosafety characteristics and the ability to degrade biogenic amines [30].

In this work, the effects of selected starters on the processing characteristics and edible safety performance of fermented mutton sausage were mainly evaluates from three aspects of protein decomposition to form free amino acids, spoilage substances (volatile base nitrogen), and biogenic amines.

## 2. Materials and Methods

### 2.1. Materials

Mutton, sheep tail fat, additives, and collagen casings were purchased in a local market. Although mutton and sheep tail fat from the same batch of mutton sheep was used, some of the meat and fat was from different animals; consequently, different bacterial species or numbers with different capacities to produce biogenic amines (BAs) might have been present. Lyophilized native starters, including *L. sakei* 3X-2B, *S. xylosus* SZ-8, and *S. carnosus* SZ-2, stored at −80 °C were used. *Lactobacillus sakei* 3X-2B were isolated from acidic Gruel, which was used because of its high acidifying ability in fermented mutton sausage [30]. Mutton and sheep tail fat come from the same batch of mutton sheep. *Staphylococcus xylosus* SZ-8 was isolated from beef jerky and selected because it could promote the decomposition of protein and formation of some fruity flavor substances in fermented mutton sausages. *Staphylococcus carnosus* SZ-2 was isolated from beef jerky, and it was used because it presented antioxidant potential in fermented mutton sausages (unpublished data). Three isolates were identified by starter safety and sequencing of the 16S ribosomal RNA gene.

### 2.2. Preparation of Dry Fermented Mutton Sausages

The following four groups were manufactured with the same process formula. Mutton sausages inoculated with single *L. sakei* 3X-2B, *L. sakei* 3X-2B + *S. xylosus* SZ-8, and mixed starters of *L. sakei* 3X-2B + *S xylosus* SZ-8 + *S. carnosus* SZ-2 were noted as the LB, LS, and LSS groups, respectively, and those without commercial starter cultures were marked as the CO group. *S. xylosus* SZ-8, *L. sakei* 3X-2B, and *S. carnosus* SZ-2 were cultured to 10^8^ CFU/mL using MSA liquid medium (Huankai Microbial Co., Guangzhou, China), MRS broth medium (Huankai Microbial Co., Guangzhou, China), and MSA liquid medium (Huankai Microbial Co., Guangzhou, China), respectively, in a shaker at 37 °C for 18 h and determined by CFU counts on their corresponding agar medium. The starter cultures were washed 2–3 times with the same amount of sterile saline solution. The mixed starter cultures were prepared on the basis of the same volume ratio. Based on the final level of 10^7^ CFU/g meat, the corresponding volume of mixed starter cultures was added to the meat. Lean meat and fat were minced and mixed with starter cultures and other ingredients. Dry fermented mutton sausage formulation included lean meat on hind legs (80% *w*/*w*) and tail fat (20% *w*/*w*), with addition of sucrose (0.5%), glucose (0.5%), NaCl (2%), and lycopene (0.5%). The fermented sausages were manufactured at the animal products processing center of Inner Mongolia Agricultural University of China. First, the mixed lean meat and fat were marinated for 12–15 h at a relative humidity (RH) of 80–85% (RH). After marinating, they were filled into collagen casings with a diameter of 20–25 mm and length of 10–15 cm and fermented for 2 days at 90% RH and 28 °C The drying and ripening cycle of sausages lasted for 6 days at 12–15 °C. During the drying and ripening period, the RH decreased everyday by 5% from 85% to 55%. Each group of 1000 g of samples was collected at 0, 2, 5, and 8 days for physical–chemical quality, proteolysis, and BAs analyses.

### 2.3. Microbial Analysis

Microbiological analyses were performed with reference to Wang et al. [29], with minor modifications. After removing the casing under aseptic conditions, 10 g of sausage was added to 90 mL of sterile saline and homogenized at 4 °C for 60 s by a homogenizer (Seward Medical, London, UK), which was counted as the first ten-fold dilution. Each time 1 mL from the previous uniform dilution was added to the next 9 mL of sterile physiological saline and mixed. Three suitable decimal dilutions were chosen and counted for the total viable counts and lactic acid bacteria, and staphylococci. The total viable counts and lactic acid bacteria were performed after cultured at 37 °C for 24 h on plate count agar (Huankai Microbial Co., Guangzhou, China) and lactic acid bacteria agar (Haibo Biological Co., Ltd., Qiangdao, China), respectively. Staphylococci was investigated at 30 °C for 24 h on mannitol salt agar (Huankai Microbial Co., Guangzhou, China). *Enterobacteriaceae* was determined after being cultured at 30 °C for 48 h on violet red bile glucose agar (Huankai Microbial Co., Guangzhou, China).

### 2.4. pH, Water Activity, Color, and TVB-N Analysis

Water activity and pH value were measured using a LabMaster-aw (Novasina AG, Schwyz, Switzerland) and digital pH meter (Mettler Toledo, Shanghai, China). The color of sausages, including lightness (L), redness (a), and yellowness (b), was observed by using a portable colorimeter (Konica Minolta CR-410, Osaka, Japan) in the CIELAB space and Lab Master-aw (Novasina AG Instruments, Switzerland). The total volatile basic nitrogen (TVB-N) was quantified by microtitration of trichloroacetic acid (TCA)—sausage extract with reference to Malle et al. [31].

### 2.5. Proteolysis Index (PI) Analysis

Proteolysis index (PI) was determined with reference to Hughes et al. [32]. Two grams of sample was added to 18 mL of distilled water, homogenized for 120 s in ice bath conditions, and centrifuged at 10,000 rpm and 4 °C for 15 min (AllegraTM, 64R, Beckman, Duarte, CA, USA). The above-mentioned solution was filtered using a Whatman 1# filter paper (Whatman, Buckinghamshire, UK). Extraction was repeated two times, and the filtrate was merged for testing. Fifteen milliliters of the above-mentioned test fluid was mixed with 15 mL of 10% TCA and then filtered with a Whatman 4# filter paper. A certain amount of filtrate was measured for nonprotein nitrogen content. The formula is as follows:$$PI(\%)=\frac{0.2\times V\times N_{1}}{N};$$

*V* indicates the merged filtrate filtered using Whatman 4#; *N*_1_ indicates the non-protein nitrogen content; *N* indicates the crude protein content in sausage.

### 2.6. Free Amino Acid (FAA) Analysis

The composition of FAA was analyzed by using an amino acid automatic analyzer (Biochrom 30+, Biochrom, Cambridge, UK). Ten grams of sausage sample was dried to constant weight at a constant temperature of 63 °C. The sample was soaked with anhydrous ether (Sinopharm Chemical Reagent Co., Shanghai, China) for 24 h, and the fat in the sample was removed by Soxhlet extraction. One gram of sample was hydrolyzed in 20 mL of 6 M HCl at 105 °C for 22–24 h. The hydrolysate was quantified with 0.1 M HCl to a 25 mL volumetric flask. More than 1 mL of hydrolysate was blown dry by nitrogen and then dissolved with 1 mL of 0.05 mol/L and filtered with a 0.45 µm filter for amino acid automatic analysis. The FAA content in sausages was expressed as mg/kg of dry matter.

### 2.7. Biogenic Amines (BAs) Analysis

According to Roseiro et al. [33], BAs in fermented mutton sausages were measured by 1260 high-performance liquid chromatography (HPLC; Agilent 1260, USA). Five grams of sausages was homogenized in ice bath and extracted with 20 mL of TCA (15% *m*/*v*). The extract was then centrifuged for 10 min at 3600 rpm, and the supernatant was filtered by using 0.45 μm filters. The above-mentioned extraction was repeated two times, and the supernatants were merged. Internal standard (1,7-diaminoheptane) was added, and the final volume was adjusted to 50 mL by using 15% TCA. The mixture of the above-mentioned extract (10 mL) and n-hexane (10 mL) was shaken for 5 min to remove fat in the extract. BAs were derivatized with dansyl chloride in alkaline medium. After removing the excess dansyl chloride, the extract was diluted by 1 mL of acetonitrile and then filtered through an Acrodisc membrane (0.45 µm; Sigma, Inc., San Francisco, CA, USA). An aliquot (20 µL) of the above-mentioned filtrate was injected into an HPLC system, and the injection was repeated three times. Chromatographic separation was conducted in a C18 Spherisorb (column specifications: 250 mm × 4.6 mm × 5 mm for the length, inner diameter, and diameter, respectively) using a gradient elution program combining ammonium acetate solution and acetonitrile (Sinopharm group, Shanghai, China). BA detection was achieved under 254 nm using an UV detector at 35 °C column temperature conditions. The final content of BAs was expressed as mg/kg in a fresh matter basis.

### 2.8. Statistical Analysis

Statistical analysis was performed using SPSS 19.0 (IBM, Chicago, IL, USA). The significance of the data was determined by one-way ANOVA. The graphs in the article were made using Origin 18.5 software.

## 3. Results and Discussion

### 3.1. Microbial Counts

Microbial changes of dry fermented sausages during fermentation and ripening with or without starter cultures are shown in Table 1. The number of the lactic acid bacteria in all groups initially increased and then decreased, which was close to the number of the total viable counts. It reached the highest on day 5 and became the dominant microbe, which was similar to that reported by Lorenzo et al. [2]. The growth potential of the lactic acid bacteria in the LSS, LS, LB, and CO groups was 4.55, 3.44, 3.10, and 3.14 log CFU/g from 0 to 8 days, respectively, and a slight decrease was observed compared with the number of the lactic acid bacteria on day 5. This result could be due to the decrease of *a*_w_ [4,30]. This also shows that there are more lactic acid bacteria in the meat. It has been reported that contaminated lactic acid bacteria may promote the formation of biological amines [22,23]. However, lactic acid bacteria, screened under certain conditions, could improve the flavor, texture, and hygiene properties of fermented sausages and inhibited pathogenic and spoilage microorganisms by acidification and production of antimicrobials [34,35]. In the whole process, the number of staphylococci in the LS and LSS groups was significantly higher than that in the CO group (*p* < 0.01), which indicated a good synergy among *L. sakei* 3X-2B, *S xylosus* SZ-8, and *S. carnosus* SZ-2. Staphylococci contributed to proteolytic and lipolytic responses, which were important to improve the color, texture, and flavor, and to inhibit lipid peroxidation of fermented meat products [36,37].

*Enterobacteriaceae* count initially increased and then decreased. *Enterobacteriaceae* count in the inoculated groups was lower than that in the CO on day 8 (the end of ripening), which might be due to the decrease of pH and growth of LAB [2,38]. The *Enterobacteriaceae* count of the LSS group was the lowest, followed by the LS group. Lorenzo et al. [2] reported that adding starter culture substantially contributed to the decrease of *Enterobacteriaceae*, which could reduce the accumulation of biogenic amines (BAs) and improve product safety quality [39].

### 3.2. pH, Water Activity, and Color

The evolution of pH, water activity (*a*_w_), and color throughout fermentation and ripening is shown in Figure 1 and Table 2. After 2 days of fermentation, the pH values in the CO, LB, LS, and LSS groups decreased from 5.9 to 5.38, 5.29, 4.93, and 4.63, respectively (Figure 1A). Based on the Pearson correlation test, pH values were negatively correlated with staphylococci and LAB counts during fermentation and ripening (r = −0.87, r = −0.90, *p* < 0.01). The increase of pH values in the LS and LSS groups after 5 to 8 days of ripening could be due to alkaline substances produced by protein decomposition, such as proteolytic amines, which were induced by bacterial proteases [31,40].

The mean values of the initial water activity (*a*_w_) of the four groups were over 0.95 (Figure 1B). The *a*_w_ of the LS and LSS groups decreased significantly faster than that of the CO group (*p* < 0.05) at days 5 and 8, which might be due to the rapid decrease in acidity of the LS and LSS groups and close to the isoelectric point of the protein, resulting in a decrease in the ability of the protein to bind water [41]. The *a*_w_ in all groups decreased to below 0.75 at day 8. Lower *a*_w_ could extend and improve the shelf life and safety of fermented sausages [33,42]. In addition, the influence of the starter on the change trend of the pH value and water activity of the sausage in this article has similar results with the previous study by Wang et al. [30] using standard strains on fermented sausage. Lightness decreased significantly (*p* < 0.01) in the four groups from initial mean value of 49.64 to 37.52 at the end of ripening (Table 2), which was positively correlated (r = 0.95, *p* < 0.01) with the decrease in *a*_w_. Redness increased rapidly for all groups during fermentation for 0–2 days (Table 2). The redness of the LSS was higher than that of the other groups. During ripening, redness showed a gradual downward trend, which might be related to the reduction of H+ concentration and the amount of nitrosoglobin formed in sausages [4].

### 3.3. Proteolysis Index (PI) and Total Volatile Basic Nitrogen (TVB-N)

Proteolysis can improve the color, texture, and flavor of sausage products [43]. Proteolysis without and with starter cultures at various stages of fermentation and ripening is shown in Figure 2A. The greatest change in proteolysis occurred between processing days 2 and 5, whereas significant differences were observed among the CO (12.81%), LB (15.10%), LS (16.77%), and LSS (16.44%) groups at the end of ripening. The results indicated that the mixed starters could promote the decomposition of proteins. Similar results were observed in silver carp sausages [44]. Nie et al. [45] and Aro et al. [46] suggested that the mixed starter cultures could promote the degradation of myofibrillar protein and sarcoplasmic protein into short peptides and small molecules. Pérez-Santaescolástica et al. [47] reported that the increase of dry-cured ham proteolysis during the processing might lead to high adhesiveness and consumer rejection of the ham. Compared with previous research by Wang et al. [29], the selected starter selected in this article has relatively moderate proteolysis ability, which was conducive to the formation of a relatively compact tissue structure of fermented lamb sausage.

The total volatile basic nitrogen (TVB-N) is a general term of alkaline metabolites such as ammonia and biogenic amines (BAs) produced by microbial decomposition of nitrogen-containing compounds such as proteins and amino acids. It is a common indicator for evaluating food quality. As shown in Figure 2B, the TVB-N value increased significantly during fermentation and ripening, and the rate of increase of TVB-N value during 0–2 days of fermentation was faster than the other stages, which was significantly and positively correlated with the growth of the protein decomposition index during fermentation (r = 0.85, *p* < 0.01). Lee et al. [48] reported that the increase in TVB-N value could be due to the combined action of microbiological and autolytic deamination of amino acids. The high bacterial producers of TVB-N primarily included *Enterobacteriaceae* spp. [49]. During 2–5 days of ripening (Figure 2B), the TVB-N value and its production rate in the LS and LSS groups were lower than those in the other groups, which indicated that the mixed starters could significantly reduce *Enterobacteriaceae*.

### 3.4. Free Amino Acid (FAA)

FAAs participated directly in taste and flavor development and changed the nutritional value of fermented sausages [50]. The FAA composition of sausages at the end of ripening is shown in Table 3. The inoculation of mixed starter cultures promoted proteolysis and increased the total FAA of fermented mutton sausages, which made the total FAA in the LS and LSS groups (224.97 and 235.53 mg/kg, respectively) significantly higher (*p* < 0.001) than the CO and LB groups (170.93 and 179.04 mg/kg, respectively). Candogan et al. [51] showed that microbial enzymes and endogenous proteases played a critical role in proteolysis and contributed to the release of FAA. Mau and Tseng [52] reported that glutamic and aspartic acids were important amine acids affecting the fresh taste; glycine and alanine could increase the sweet taste of food, whereas phenylalanine, lysine, leucine, valine, and arginine could result in the bitter taste of food. The total content of glutamic and aspartic acids in the LS and LSS groups (41.44 and 38.34 mg/kg, respectively) were higher than those in the CO and LB groups (30.97 and 33.32 mg/kg, respectively). In addition, the total content of glycine and alanine in the LS and LSS groups (22.35 and 21.23 mg/kg, respectively) were higher than those in the CO and LB groups (18.37 and 18.34 mg/kg, respectively). This result indicated that the mixed starter could improve the fresh taste and sweet taste of the fermented sausages. Other FAAs showed sour or salty characteristics. The accumulation of these amino acids in sausages was of great significance for improving the nutritional quality, taste, and flavor of sausages. The key aroma compounds (isovaleraldehyde, isoamyl alcohol, isovaleric acid, methional, methanethiol, dimethyl trisulfide, phenylacetic acid, and phenylacetaldehyde) identified in Swiss-type cheese primarily resulted from the catabolism of branched-chain amino acids, methionine, and phenylalanine [53,54,55]. According to Stahnke [3], the branched-chain amino acids could be metabolized by starter culture to generate isovaleraldehyde and isoamyl alcohol with fruity aroma, which were characteristic flavor substances of fermented meat. Isovaleraldehyde (3-methyl butyraldehyde) reacting with sulfur compounds could generate a flavor similar to bacon [56]. The mixed starter cultures increased the content of isoleucine and leucine in the LS (11.74 and 16.37 mg/kg, respectively) and LSS groups (11.50 and 17.34 mg/kg, respectively), which was higher than those in the CO and LB. Therefore, *L. sakei* 3X-2B + *S. xylosus* SZ-8 and *L. sakei* 3X-2B + *S. xylosus* SZ-8 + *S. carnosus* SZ-2 could increase the content of flavor precursors and flavor substances in sausages.

Some FAA had higher content, and they were considered as the predominant amino acids in the final mixes, such as leucine, arginine, and glutamate, which were higher in the LSS and LS groups than in the CO and LS groups. This result showed that *L. sakei* 3X-2B + *S. xylosus* SZ-8 and *L. sakei* 3X-2B + *S. xylosus* SZ-8 + *S. carnosus* SZ-2 had a proteolytic activity. As reported by Hughes et al. [32] and Casaburi et al. [37], endogenous enzymes in meat and microbial enzymes played a crucial role in proteolysis during processing of sausages. However, compared with the previous research results of Wang et al. [29] using standard strains on the free amino acid composition of fermented mutton sausages, the starters selected in this article were more conducive to the formation and accumulation of FAA.

### 3.5. Biogenic Amines (BAs) Analysis

Biogenic amines (BAs) are low-molecular-weight compounds with biological activity and amino group, and they are primarily generated by the action of microbial amino acid decarboxylase on amino acid decarboxylation [57,58]. The accumulation of BAs in dry fermented mutton sausages during processing was determined by HPLC (Agilent 1260; Table 4). The gradual increase in total BAs content, particularly in the CO group, was primarily related to the changes of TYR, PUT, and HIS, which was similar to that reported by Lu et al. [59] and Domínguez et al. [38]. The total BAs in sausages increased significantly from approximately 6.80 mg/kg on average at 0 day to 219.50, 129.31, 95.12, and 99.05 mg/kg for the CO, LB, LS, and LSS groups after 8 days of ripening, respectively. The total content of BAs was lower than the maximum limit of 1000 mg/kg, which was considered detrimental to human health [60]. As shown in Table 4, the inoculated groups were lower than the CO group, which indicated that the starter cultures mixed with *L. sakei* 3X-2B + *S. xylosus* SZ-8 and *L. sakei* 3X-2B + *S. xylosus* SZ-8 + *S. carnosus* SZ-2 could effectively inhibit the accumulation of BAs in the fermented mutton sausages. Based on previous reports, *Pseudomonas*, *Enterobacteriaceae*, and Lactobacilli were the main producers of BAs [4,16]. Our research showed that mixed starters could reduce the number of *Enterobacteriaceae* in sausages, which was in agreement with the changes of BAs in the inoculated group. The content of CAD, PUT, and HIS in the LS and LSS groups showed an opposite trend to the increase of the corresponding precursor amino acid (lysine, arginine, and histidine) content. Furthermore, applying low temperature and strong acidification to starters during processing did not favor BA formation, which could also be explained by inhibiting the growth of spoilage microorganisms and FAA decarboxylase activity [61].

Compared to day 0, the content of tryptamine (TRY) in the four groups CO, LB, LS, and LSS increased by 9.21, 5.11, 5.39, and 4.82, respectively, at the end of ripening. The concentration of TRY in the inoculated groups was lower (*p* < 0.001) than that in the CO group, whereas the TRY change of the three groups inoculated with different mixed starters showed no significant difference. This result indicated that the inoculation of *L. sakei* inhibited decarboxylase activity and resulted in the formation of TRY in fermented mutton sausages, which was in agreement with the reports by Sun et al. [62]. At the end of ripening, the TRY content of the three groups inoculated with starters was lower than that of the CO. According to Güven et al. [63], PHE was an aromatic amine that can inhibit the diamine oxidase activity, which was metabolized by monoamine oxidase. The amount of PHE in the CO, LB, LS, and LSS groups increased by 2.38, 2.79, 2.29, and 1.92 mg/kg at the end of ripening, respectively. PHE in the LS and LSS groups was lower than that in the CO and LB groups. This result indicated that the mixed starters could effectively prevent the formation of PHE.

CAD and PUT were often used as an important indicator of red meat and food hygiene [64]. CAD and PUT in sausages were lower at day 0 (Table 4). However, PUT in sausages increased significantly faster than CAD during processing. Compared with the CO group, PUT of the inoculated groups was lower at day 8. However, the difference between the LS and LSS groups was not significant (*p* > 0.05), and the LSS group was the lowest. Latorre-Moratalla et al. [65] and Suzzi and Gardini [66] reported that high CAD levels in fermented sausages were related to high amino acid decarboxylase activity and *Enterobacteriaceae* count. In our study, inoculation of mixed starter cultures inhibited the counts of *Enterobacteriaceae* and the accumulation of CAD and PUT, which indicated that the mixed starters could improve hygiene quality of fermented dry mutton sausages. The total content of CAD and PUT in the CO, LB, LS, and LSS groups was 31.93, 19.42, 6.95, and 6.84 mg/kg, respectively. The total content of CAD and PUT in the LS, and LSS groups were significantly lower than those in the CO and LB groups, which indicated that the addition of the mixed starters could effectively inhibit the number of *Enterobacteriaceae*. Nie et al. [45] reported that starter culture mixed with *L. plantarum* ZY40 plus *S. cerevisiae* JM19 significantly inhibited and reduced the PUT and CAD by more than 37% and 76%, respectively. Bover-Cid et al. [67] reported similar results.

HIS is the most toxic biological amine, and the level of its toxicity is related to the absorption concentration and presence of other amines, such as CAD, PUT, and TYR [15,16]. HIS in the CO and LS groups increased during ripening, and the increase in the CO group was significantly (*p* < 0.001) higher than that in the inoculated groups. At the end of ripening, HIS in the inoculated groups (17.30, 4.30, and 4.39 mg/kg in the LB, LS, and LSS groups relative to day 0, respectively) was significantly lower than that in the CO group (29.36 mg/kg). However, the content of HIS in sausages was lower than its tolerance level according to the US Food and Drug Administration, which was 100 mg/kg in flesh fish and 50 mg/kg as a guidance level (FDA, 1990, Rockwell, Milwaukee, WI, USA). Compared with the inoculated groups, the above-mentioned results showed that adding mixed starter cultures could inhibit the accumulation of HIS effectively. The content of HIS in the LS and LSS groups was lower than the mean values reported by EFSA (2011): European sausages (approximately 25 mg/kg) and Harbin dry sausages [62]. TYR is the second important BA, and its toxicological level ranges from 150 mg/kg to 800 mg/kg. However, TYR was the most detected biological amine among the four groups of sausages. As shown in Table 4, the increase of TYR primarily occurred during 0–5 days of fermentation and early ripening. Furthermore, the differences in the TYR content between the inoculated and CO groups were primarily found at the end of ripening.

## 4. Conclusions

Inoculation of the mixed starter cultures (*L. sakei* 3X-2B + *S. xylosus* SZ-8 + *S. carnosus* SZ-2) increased the number of lactic acid bacteria and staphylococci, which accelerated acidification and decreased *a*_w_. The mixed starter cultures contributed to proteolysis and increased the total FAAs. However, the mixed starter cultures inhibited the reproduction of *Enterobacteriaceae* and accumulation of BAs. The accumulation of BAs in sausages showed an opposite trend to the increase of amino acids, which achieved our expectation for *L. sakei* 3X-2B + *S. xylosus* SZ-8 + *S. carnosus* SZ-2-reducing BAs. Sausages made with the mixed starter cultures are safe for consumers. Therefore, *L. sakei* 3X-2B + *S. xylosus* SZ-8 + *S. carnosus* SZ-2 was considered as potential starters.

## Figures and Tables

**Figure 1 foods-10-02939-f001:**
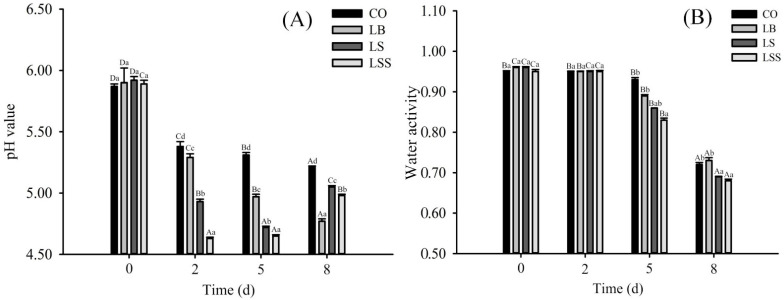
Effect of mixed starter cultures on pH value (**A**) and water activity (*a*_w_) (**B**) of dry fermented mutton sausage at fermentation and ripening stages. ^a,b,c,d^: Mean values followed different lowercase letters in the same days of ripening indicate significant difference (*p* < 0.05). ^A,B,C,D^: Mean values followed different uppercase letters in the same batch indicate significant difference (*p* < 0.05). CO: without starter culture; LB: *L. sakei* 3X-2B; LS: *L. sakei* 3X-2B + *S. xylosus* SZ-8; LSS: *L. sakei* 3X-2B + *S xylosus* SZ-8 + *S. carnosus* SZ-2.

**Figure 2 foods-10-02939-f002:**
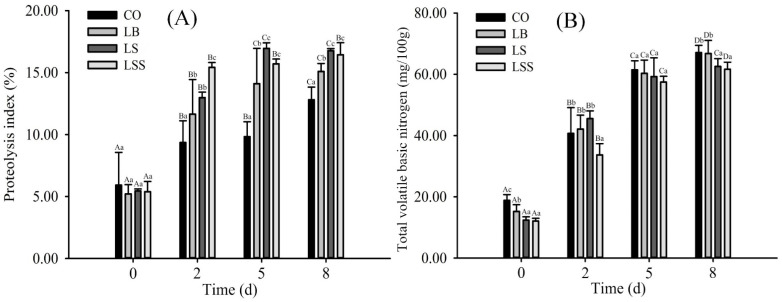
Effect of mixed starter cultures on protein decomposing index (**A**) and total volatile basic nitrogen (**B**) of dry fermented mutton sausage at fermentation and ripening stages. ^a,b,c^: Mean values followed different lowercase letters in the same days of ripening indicate significant difference (*p* < 0.05). ^A,B,C,D^: Mean values followed different uppercase letters in the same batch indicate significant difference (*p* < 0.05).

**Table 1 foods-10-02939-t001:** Effect of mixed starter cultures on different microbial groups of dry fermented sausages at fermentation and ripening stages.

	Days	Batch				Sign.
		CO	LB	LS	LSS	
Total viable counts (Log CFU/g)	0	5.55 ± 0.02 ^Aa^	6.55 ± 0.04 ^Ab^	6.45 ± 0.13 ^Ab^	6.50 ± 0.10 ^Ab^	*
	2	9.50 ± 0.02 ^Ba^	9.57 ± 0.01 ^Ba^	9.62 ± 0.10 ^Ba^	9.60 ± 0.14 ^Ba^	n.s.
	5	9.35 ± 0.05 ^Ba^	9.16 ± 0.02 ^Ba^	9.77 ± 0.07 ^Ba^	9.97 ± 0.05 ^Ba^	n.s.
	8	9.21 ± 0.08 ^Ba^	9.27 ± 0.10 ^Ba^	9.35 ± 0.05 ^Ba^	9.92 ± 0.15 ^Bb^	*
	Sign.	***	***	***	***	
Lactic acid bacteria (Log CFU/g)	0	4.60 ± 0.30 ^Aa^	5.74 ± 0.21 ^Ab^	5.71 ± 0.07 ^Ab^	5.80 ± 0.07 ^Ab^	***
	2	8.47 ± 0.27 ^Ba^	8.48 ± 0.05 ^Ba^	8.59 ± 0.03 ^Ba^	9.00 ± 0.04 ^Bb^	**
	5	9.21 ± 0.25 ^Ca^	9.20 ± 0.21 ^Ca^	9.09 ± 0.12 ^Da^	9.80 ± 0.92 ^Ba^	n.s.
	8	9.15 ± 0.31 ^Ca^	9.18 ± 0.12 ^Ca^	8.81 ± 0.12 ^Ca^	8.94 ± 0.67 ^Ba^	n.s.
	Sign.	***	***	***	***	
Staphylococci (Log CFU/g)	0	3.23 ± 0.25 ^Aa^	5.25 ± 0.02 ^Ab^	5.37 ± 0.02 ^Ab^	5.55 ± 0.03 ^Ab^	***
	2	5.69 ± 0.01 ^Ba^	6.92 ± 0.25 ^Bb^	7.28 ± 0.25 ^Bb^	8.03 ± 0.26 ^Bc^	**
	5	7.84 ± 0.08 ^Cb^	7.14 ± 0.03 ^Ba^	8.74 ± 0.23 ^Bc^	8.89 ± 0.01 ^Cc^	***
	8	7.22 ± 0.30 ^Ca^	8.55 ± 0.18 ^Cb^	9.31 ± 0.15 ^Cc^	9.37 ± 0.21 ^Cc^	***
	Sign.	***	***	***	***	
*Enterobacteriaceae* (Log CFU/g)	0	3.28 ± 0.07 ^Ba^	3.29 ± 0.02 ^Ca^	3.51 ± 0.08 ^Bb^	3.47 ± 0.08 ^Bb^	**
	2	4.78 ± 0.02 ^Cb^	4.78 ± 0.03 ^Db^	4.30 ± 0.05 ^Ca^	4.89 ± 0.04 ^Cb^	***
	5	3.50 ± 0.24 ^Bc^	2.93 ± 0.02 ^Bb^	2.64 ± 0.07 ^Aa^	3.11 ± 0.18 ^Bb^	**
	8	2.99 ± 0.16 ^Ac^	2.36 ± 0.12 ^Ab^	2.35 ± 0.35 ^Ab^	1.74 ± 0.44 ^Aa^	***
	Sign.	***	***	***	***	

^a,b,c^: Mean values followed different lowercase letters in the same row indicate significant difference. ^A,B,C,D^: Mean values followed different uppercase letter in the same column indicate significant difference. Sign.: significance; n.s.: not significant; * (*p* < 0.05); ** (*p* < 0.01); *** (*p* < 0.001). CO: without starter culture; LB: *L. sakei* 3X-2B; LS: *L. sakei* 3X-2B + *S. xylosus* SZ-8; LSS: *L. sakei* 3X-2B + *S xylosus* SZ-8 + *S. carnosus* SZ-2.

**Table 2 foods-10-02939-t002:** Effect of mixed starter cultures on color of dry fermented mutton sausage at fermentation and ripening stages.

Colour	Days	Batch				Sign.
		CO	LB	LS	LSS	
Lightness (L)	0	49.21 ± 1.10 ^Ba^	51.52 ± 2.03 ^Ba^	49.46 ± 1.04 ^Ba^	48.37 ± 2.11 ^Ba^	n.s.
	2	48.96 ± 0.14 ^Ba^	49.26 ± 0.39 ^Ba^	49.76 ± 0.20 ^Ba^	48.09 ± 4.66 ^Ba^	n.s.
	5	37.13 ± 0.11 ^Aa^	36.52 ± 1.87 ^Aa^	36.24 ± 0.59 ^Aa^	35.01 ± 4.50 ^Aa^	n.s.
	8	37.59 ± 0.12 ^Aa^	37.34 ± 0.12 ^Aa^	35.63 ± 0.06 ^Aa^	39.53 ± 3.89 ^Aa^	n.s.
	Sign.	***	***	***	***	
Redness (a)	0	10.19 ± 0.03 ^Aa^	10.22 ± 0.17 ^Aa^	9.86 ± 0.10 ^Aa^	8.43 ± 0.59 ^Aa^	n.s.
	2	18.65 ± 0.06 ^Da^	19.99 ± 0.28 ^Ba^	20.76 ± 0.18 ^Ba^	26.05 ± 3.74 ^Bb^	**
	5	15.36 ± 0.12 ^Ba^	21.78 ± 0.86 ^Bb^	22.45 ± 0.14 ^Bb^	25.47 ± 5.33 ^Bb^	***
	8	17.37 ± 0.31 ^Ca^	20.16 ± 0.20 ^Bb^	22.48 ± 0.04 ^Bc^	21.53 ± 2.04 ^Bbc^	***
	Sign.	***	***	***	***	
Yellowness (b)	0	14.50 ± 0.24 ^Da^	14.41 ± 0.47 ^Da^	13.60 ± 0.18 ^Ba^	14.31 ± 0.94 ^Aa^	***
	2	10.64 ± 0.43 ^Ca^	12.32 ± 0.19 ^Cb^	10.44 ± 0.18 ^Aa^	12.48 ± 0.40 ^Cb^	**
	5	6.62 ± 0.03 ^Aa^	10.53 ± 1.04 ^Bb^	9.52 ± 0.04 ^Ab^	11.12 ± 0.97 ^Bb^	n.s.
	8	8.42 ± 0.20 ^Ba^	8.67 ± 0.06 ^Aa^	10.12 ± 0.11 ^Ab^	11.20 ± 0.51 ^Bb^	n.s.
	Sign.	**	**	**	**	

^a,b,c^: Mean values followed different lowercase letters in the same row indicate significant difference. ^A,B,C,D^: Mean values followed different uppercase letter in the same column indicate significant difference. Sign.: significance; n.s.: not significant; ** (*p* < 0.01); *** (*p* < 0.001).

**Table 3 foods-10-02939-t003:** Effect of different mixed starter cultures on free amino acids (FAA) (expressed as mg/kg of dry matter) of dry fermented mutton sausage at the end of ripening.

FAA	Batch				Sign.
	CO	LB	LS	LSS	
Aspartic acid	10.65 ± 3.37 ^a^	9.89 ± 0.27 ^a^	14.08 ± 2.12 ^a^	13.60 ± 3.01 ^a^	n.s.
Threonine	8.48 ± 1.27 ^a^	9.51 ± 0.23 ^a^	11.28 ± 2.96 ^a^	12.87 ± 1.84 ^a^	n.s.
Serine	7.52 ± 1.05 ^a^	8.61 ± 0.19 ^a^	10.13 ± 2.64 ^a^	9.84 ± 0.57 ^a^	n.s.
Glutamic acid	20.32 ± 2.68 ^a^	23.43 ± 0.47 ^a^	27.36 ± 6.87 ^a^	24.74 ± 1.42 ^a^	n.s.
Glycine	8.94 ± 0.82 ^a^	9.60 ± 0.08 ^a^	9.64 ± 0.03 ^a^	9.37 ± 1.15 ^a^	n.s.
Alanine	9.43 ± 1.44 ^a^	8.74 ± 1.82 ^a^	12.71 ± 2.92 ^a^	11.86 ± 3.22 ^a^	n.s.
Cystine	13.43 ± 1.14 ^b^	6.54 ± 1.96 ^a^	19.71 ± 5.58 ^c^	16.72 ± 0.30 ^c^	***
Valine	7.51 ± 0.90 ^a^	8.81 ± 0.19 ^a^	10.06 ± 2.63 ^a^	9.50 ± 1.66 ^a^	n.s.
Methionine	5.91 ± 1.02 ^a^	6.25 ± 0.23 ^a^	7.36 ± 2.97 ^ab^	8.15 ± 0.63 ^b^	*
Isoleucine	9.41 ± 1.25 ^a^	10.30 ± 0.37 ^a^	11.74 ± 4.28 ^a^	11.50 ± 1.57 ^a^	n.s.
Leucine	12.23 ± 1.43 ^a^	14.26 ± 0.32 ^ab^	16.37 ± 4.16 ^ab^	17.34 ± 0.93 ^b^	*
Tyrosine	7.80 ± 1.32 ^a^	7.97 ± 0.28 ^a^	9.53 ± 3.82 ^ab^	10.22 ± 0.07 ^b^	*
Phenylalanine	9.22 ± 1.55 ^a^	9.49 ± 0.33 ^a^	11.25 ± 4.47 ^a^	11.64 ± 2.15 ^a^	n.s.
Lysine	7.31 ± 0.95 ^a^	8.58 ± 0.16 ^a^	9.90 ± 2.49 ^a^	10.34 ± 1.99 ^a^	n.s.
Histidine	6.77 ± 0.88 ^a^	7.93 ± 0.19 ^a^	9.29 ± 2.45 ^a^	18.82 ± 1.52 ^b^	***
Arginine	11.30 ± 1.69 ^a^	13.45 ± 0.44 ^a^	15.72 ± 4.65 ^ab^	21.93 ± 3.61 ^b^	***
Proline	14.70 ± 2.47 ^a^	15.68 ± 0.35 ^a^	18.85 ± 7.31 ^a^	17.09 ± 0.75 ^a^	n.s.
Total free amino acids	170.93 ± 3.57 ^a^	179.04 ± 4.18 ^a^	224.97 ± 5.05 ^b^	235.53 ± 4.79 ^b^	***

^a,b,c^: Mean values followed different lowercase letters in the same row indicate significant difference. (*p* < 0.05). Sign.: significance; n.s.: not significant; * (*p* < 0.05); *** (*p* < 0.001).

**Table 4 foods-10-02939-t004:** Effect of mixed starter cultures on biogenic amines of dry fermented mutton sausage at fermentation and ripening stages.

BAs (mg/kg)	Days	Batch				Sign.
		CO	LB	LS	LSS	
TRY	0	1.45 ± 0.54 ^Aa^	0.93 ± 0.19 ^Aa^	1.16 ± 0.27 ^Aa^	1.43 ± 0.32 ^Aa^	n.s.
	2	3.64 ± 0.29 ^Ba^	2.98 ± 0.56 ^Ba^	3.15 ± 0.47 ^Ba^	3.46 ± 0.37 ^Ba^	n.s.
	5	6.72 ± 0.73 ^Cab^	4.79 ± 0.78 ^Ca^	7.77 ± 1.88 ^Cb^	6.59 ± 0.46 ^Cab^	*
	8	10.66 ± 1.81 ^Db^	6.04 ± 0.69 ^Da^	6.56 ± 0.49 ^Ca^	6.25 ± 0.21 ^Ca^	***
	Sign.	***	***	***	***	
PHE	0	1.25 ± 0.04 ^Ac^	0.45 ± 0.06 ^Aa^	0.44 ± 0.06 ^Aa^	0.84 ± 0.17 ^Ab^	***
	2	1.52 ± 0.24 ^Ab^	1.36 ± 0.26A ^Bb^	0.96 ± 0.23 ^Aa^	0.95 ± 0.13 ^Aa^	*
	5	1.64 ± 0.76 ^Aa^	2.61 ± 0.57 ^Cab^	2.48 ± 0.78 ^Bab^	2.91 ± 0.02 ^Bb^	*
	8	3.63 ± 1.30 ^Ba^	3.24 ± 0.20 ^Da^	2.73 ± 0.32 ^Ba^	2.72 ± 0.34 ^Ba^	n.s.
	Sign.	**	***	***		
PUT	0	1.79 ± 0.69 ^Aa^	1.57 ± 0.19 ^Aa^	1.29 ± 0.20 ^Aa^	1.51 ± 0.07 ^Aa^	n.s.
	2	18.91 ± 5.62 ^Bb^	4.82 ± 0.63 ^Ba^	3.23 ± 0.84 ^Ba^	3.82 ± 0.21 ^Ba^	***
	5	48.38 ± 9.05 ^Cb^	11.53 ± 2.15 ^Ca^	7.12 ± 1.50 ^Ca^	6.57 ± 0.89 ^Ca^	***
	8	62.87 ± 5.97 ^Dc^	16.11 ± 2.25 ^Db^	6.09 ± 0.77 ^Ca^	5.72 ± 1.08 ^Ca^	***
	Sign.	***	***	***	***	
CAD	0	0.15 ± 0.03 ^Aa^	0.13 ± 0.02 ^Aa^	0.12 ± 0.05 ^Aa^	0.17 ± 0.05 ^Aa^	n.s.
	2	1.52 ± 0.58 ^Bb^	0.59 ± 0.08 ^Ba^	0.36 ± 0.08 ^Aa^	0.51 ± 0.15 ^Ba^	**
	5	1.19 ± 0.15 ^Ba^	1.66 ± 0.31 ^Da^	1.11 ± 0.28 ^Ba^	1.24 ± 0.39 ^Ca^	n.s.
	8	1.63 ± 0.19 ^Bc^	1.25 ± 0.13 ^Cb^	0.86 ± 0.04 ^Ba^	1.12 ± 0.14 ^Cb^	***
	Sign.	**	***	***	***	
HIS	0	0.94 ± 0.11 ^Aa^	0.87 ± 0.20 ^Aa^	0.87 ± 0.16 ^Aa^	1.17 ± 0.22 ^Aa^	n.s.
	2	16.14 ± 5.44 ^Bb^	8.67 ± 0.13 ^Ab^	6.20 ± 0.78 ^Ab^	6.93 ± 1.04 ^Ab^	**
	5	13.99 ± 2.60 ^Bb^	14.42 ± 1.10 ^Bc^	9.65 ± 1.79 ^Ac^	10.09 ± 0.95 ^Ac^	**
	8	30.30 ± 1.51 ^Cc^	18.17 ± 2.13 ^Bd^	5.17 ± 0.70 ^Ab^	5.56 ± 0.29 ^Ab^	***
	Sign.	***	***	***	***	
TYR	0	2.49 ± 0.06 ^Ba^	2.01 ± 0.08 ^ABa^	2.69 ± 0.88 ^Ba^	1.17 ± 0.24 ^Aa^	*
	2	21.98 ± 1.47 ^Ab^	34.55 ± 6.88 ^Bb^	25.35 ± 6.59 ^ABb^	26.55 ± 5.36 ^ABb^	*
	5	77.69 ± 6.45 ^Ac^	71.37 ± 9.98 ^Ac^	76.35 ± 4.86 ^Ac^	74.45 ± 5.30 ^Ac^	n.s.
	8	110.40 ± 5.49 ^Bd^	84.50 ± 8.82 ^Ac^	73.70 ± 4.08 ^Ac^	77.69 ± 5.48 ^Ac^	***
	Sign.	***	***	***	***	

^a,b,c,d^: Mean values followed different lowercase letters in the same column indicate significant difference. ^A,B,C,D^: Mean values followed different uppercase letter in the same row indicate significant difference. Significance: n.s.: not significant; * (*p* < 0.05); ** (*p* < 0.01); *** (*p* < 0.001). BAs: Biogenic amines; TRY: tryptamine; PHE: phenylethylamine; PUT: putrescine; CAD: cadaverine; TYR: tyramine; HIS: histamine.

## Data Availability

Not applicable.

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
