# Peer review of "Effect of Mixed Starters on Proteolysis and Formation of Biogenic Amines in Dry Fermented Mutton Sausages"

_foods, 2021, doi:10.3390/foods10122939_

Round 1

Reviewer 1 Report

I think that with the changes made after the first review now the differences between the other paper are more clear. Moreover the results are written better than before, so the reader can appreciate the effects of the selected strains.

Author Response

Thank you for your guidance.

Reviewer 2 Report

Comments and Suggestions for Authors

General comments

The authors compared the effects of different starter cultures as single cultures or mixed starters on the commensal microbiota, inhibition of biogenic amine formation, and product characteristics in dry fermented mutton sausage. They showed that mixed starter cultures have antimicrobial activity against commensal microbiota and reduced the production of BAs.  Suggesting that mixed starter cultures have potential application in fermented meat products, improving their safety.

Strengths:

BAs in sausages represent a health risk for consumers, thus investigating the BAs accumulation mechanisms and ways to inhibit them is important to improve fermented sausage safety. The study is thought-provoking and adds more understanding and options to limit BA accumulation in fermented sausages. The study also helps in the identification and development of candidates for BAs control in fermented meat products. The use of several approaches to try and establish the benefits of mixed starter cultures is a great plus for the paper.

Suggestions to the authors:

The paper still needs strong language editing as some points are not coming out clearly, which detracts from the overall good work done. In the introduction, the section on potential negative health effects of BAs needs to be summarised and requires language editing.

My main concerns are about how materials and methods are described and, consequently, about how results are displayed, interpreted, and discussed. Some details are missing and should be specified. For instance, it’s not clear if meat or fat from the same animal was used in the preparation of all the sausages. If it was not from the same animal, it might have different commensal microbiota with differing capabilities of BAs production. This needs to be clearly defined. If meat or fat from different animals was used, a statement that such data must be interpreted with caution due to this weakness needs to be included.

More importantly, a similar project was done and is published by the authors group.

  1. Wang, D.B.; Zhao, L.H.; Su, R.N.; Jin, Y. Effects of different starter culture combinations on microbial counts and physico-chemical properties in dry fermented mutton sausages. Food Science Nutrition, 2019, 7, 1957-1968. 496.

You used the same strains, set-up, and came to the same conclusions except for the new data on BAs. In the current paper you only reference this work in passing in the methods line 100 and results line 218. I recommend that you restructure your article to highlight this previous work and discuss your current findings also referencing this previous work. If this was a follow up study, please report it as such, focusing more on the new findings as it might seem as though you are trying to republish the same work twice. The new information on BAs is still very relevant and has merit.

Data analysis: Because the starting amounts of BAs or CFU counts of commensal microbiota were different in some samples, the authors should calculate the growth potential observed for each treatment or relative changes to each value for the analysis to account for the initial variation of each sample (e.g., “day 0 - day 8 CFU counts” or “day8 BA/day0 BA”). For instance, in Table 4, PHE of CO on day 0 was 1.25±0.04 compared to 0.45±0.06 on LB, 0.44±0.06 on LS, and 0.84±0.17 on LSS. Relative change factors are CO (2.904), LB (7.2), LS (6.205), and LSS (3.238). Meaning that relative to starting value of PHE on day 0, CO had the least increase in this BA.

Results and discussion. Some results need to be analysed and interpreted using relative comparison to day 0 values as highlighted above. Such analysis will necessitate some changes to the results and discussion sections. It would be prudent for the authors to highlight their study setup limitations.

Specific comments

Abstract

Line 35: please consider rephrasing to “This study showed that L. sakei 3X-2B + S. xylosus SZ-8 + S. carnosus SZ-2 is a potential mixed starter for fermented meat products.”

Introduction

The introduction needs language editing as some points are not coming out clearly which detracts from the overall good work done in generating it.

Line 42: please rephrase to “are formed due to”

Line 51-63: Please summarise and consider language editing this section.

Line 62: please rephrase: “BAs are known as potential precursor….”.

Line 91-93: A similar project was done and is published by your group. You used the same strains, set-up, and came to the same conclusions except for the new data on BAs. In the current paper you only reference this work in passing line 100 and 218. I recommend you restructure your article to highlight this previous work and discuss your current findings also referencing this previous work. If this was a follow up study, please focus more on the new findings as it might seem as though you are trying to republish the same work twice. The new information on BAs is still very relevant and has merit.

Methods

Line 96: It is not clear if meat or fat from the same animal was used in the preparation of all the sausages. If it was not from the same animal, it might have different commensal microbiota with differing capabilities of BAs production. This needs to be clearly defined. If meat or fat from different animals was used then a statement that such data must be interpreted with caution due to this weakness needs to be included.

Line 104-105:  How was the starter safety determined especially if whole-genome sequencing was not done? For practical application on products to be consumed, is 16S rRNA sequencing only sufficient to determine safety for such starter cultures?

Line 112: what is MRS and MSA. Who are the manufactures, country, city, etc?

Line 113-114: consider rephrasing to "and determined by CFU counts on their corresponding agar medium".

Line 138: Were the LBS Agar plates incubated anaerobically or aerobically and what was the duration of incubation? I am not familiar with this LBS agar, can you provide the reviewer with more information on this media as a quick look for it online was not successful.

Line 211-231: Please use growth potential for comparison not just numbers as the starting CFUs of each species on each treatment differed on day 0. CFU change factor or growth potential (e.g., 1 log CFU vs 2 log CFU as arbitrary examples) is more valuable for interpretation rather than the final CFU numbers themselves alone. For instance, the growth potential of LAB was greater on CO compared to LB, LS, and LSS, probably due to the absence of competition from the starter cultures. Please comment on the “Jamerson effect” in relation to your observations.

Line 247-251: please review this statement on aw and support it with more appropriate references.

Line 261: Table 2: What could be the reason for the significant difference in lightness of LB on day 5?

Line 414: How accurate are interpretations from this table if the relative changes to these BAs was not considered.

Table 4, PHE of CO on day 0 was 1.25±0.04 compared to 0.45±0.06 on LB, 0.44±0.06 on LS, and 0.84±0.17 on LSS. Relative change factors are CO (2.904), LB (7.2), LS (6.205), and LSS (3.238). Meaning that relative to starting value of PHE on day 0, CO had the least increase in this BA. A similar effect might be observed on TRY and TYR.

Please calculate the relative changes for all the other BAs and then make the appropriate comments on inhibition of BAs production.

References:

Reference 39 needs to be corrected.

Author Response

Specific comments

Abstract

Line 35: please consider rephrasing to “This study showed that L. sakei 3X-2B + S. xylosus SZ-8 + S. carnosus SZ-2 is a potential mixed starter for fermented meat products.”

Response: Thank you for your guidance. According to your suggestion, we have rephrased. 

Introduction

The introduction needs language editing as some points are not coming out clearly which detracts from the overall good work done in generating it.

Line 42: please rephrase to “are formed due to”

Response: Thank you for your guidance. According to your suggestion, we have rephrased (line 42).

Line 51-63: Please summarise and consider language editing this section.

Response: Thank you for your guidance. we have revised.

Line 62: please rephrase: “BAs are known as potential precursor….”.

Response: Thank you for your guidance. According to your suggestion, we have rephrased (line 64).

Line 91-93: A similar project was done and is published by your group. You used the same strains, set-up, and came to the same conclusions except for the new data on BAs. In the current paper you only reference this work in passing line 100 and 218. I recommend you restructure your article to highlight this previous work and discuss your current findings also referencing this previous work. If this was a follow up study, please focus more on the new findings as it might seem as though you are trying to republish the same work twice. The new information on BAs is still very relevant and has merit.

Response: Thank you for your guidance. we have revised.

Methods

Line 96: It is not clear if meat or fat from the same animal was used in the preparation of all the sausages. If it was not from the same animal, it might have different commensal microbiota with differing capabilities of BAs production. This needs to be clearly defined. If meat or fat from different animals was used then a statement that such data must be interpreted with caution due to this weakness needs to be included.

Response: Thank you for your guidance. The raw meat of all the sausages made in this article are from the same sheep. Because these hind legs and sheep tails are purchased in the slaughter market. And it has been explained in the text.

Line 104-105:  How was the starter safety determined especially if whole-genome sequencing was not done? For practical application on products to be consumed, is 16S rRNA sequencing only sufficient to determine safety for such starter cultures?

Response: Thank you for your question, we have responded to the safety of the strain last time and clearly explained it. We have identified safety of the starters used in this article by hemolysis and thermostable nuclease test in previous studies.

Line 112: what is MRS and MSA. Who are the manufactures, country, city, etc?

Response: Thank you for your question. We have added the information of the manufactures, country, city, etc (lines 116-118).

Line 113-114: consider rephrasing to "and determined by CFU counts on their corresponding agar medium".

Response: Thank you for your guidance. According to your suggestion, we have rephrased (line 119).

Line 138: Were the LBS Agar plates incubated anaerobically or aerobically and what was the duration of incubation? I am not familiar with this LBS agar, can you provide the reviewer with more information on this media as a quick look for it online was not successful.

Response: LBS agar medium was purchased from Qingdao Haibo Biological Co., Ltd. LBS agar medium should be placed in 36±1℃ for aerobic culture for 24-48 hours when measuring lactic acid bacteria.

Line 211-231: Please use growth potential for comparison not just numbers as the starting CFUs of each species on each treatment differed on day 0. CFU change factor or growth potential (e.g., 1 log CFU vs 2 log CFU as arbitrary examples) is more valuable for interpretation rather than the final CFU numbers themselves alone. For instance, the growth potential of LAB was greater on CO compared to LB, LS, and LSS, probably due to the absence of competition from the starter cultures. Please comment on the “Jamerson effect in relation to your observations.

Response: Thank you for your guidance. According to your suggestion, we have rephrased (lines 221-227). Sorry, the “Jamerson effect” is not mentioned in this manuscript.

Line 247-251: please review this statement on aw and support it with more appropriate references.

Response: Thank you for your guidance. We have added (line 257).

Line 261: Table 2: What could be the reason for the significant difference in lightness of LB on day 5?

Response: Thank you for your guidance. After inspection, there is no significant difference between them. This is an error in our input data, which has been corrected.

Line 414: How accurate are interpretations from this table if the relative changes to these BAs was not considered.

Table 4, PHE of CO on day 0 was 1.25±0.04 compared to 0.45±0.06 on LB, 0.44±0.06 on LS, and 0.84±0.17 on LSS. Relative change factors are CO (2.904), LB (7.2), LS (6.205), and LSS (3.238). Meaning that relative to starting value of PHE on day 0, CO had the least increase in this BA. A similar effect might be observed on TRY and TYR.

Please calculate the relative changes for all the other BAs and then make the appropriate comments on inhibition of BAs production.

Response: Thank you for your question. For histamine and tyramine, the most toxic to biogenic amines, the increase in the control group were much higher than those in the starter groups. Among all the data, there is only one, and in the early stage of fermentation and maturation, the content of phenethylamine in the starter groups are higher than that in the control group. Therefore, the content change analysis used in this article is accurate.

References:

Reference 39 needs to be corrected.

Response: Thank you for your guidance. We have corrected.

Reviewer 3 Report

The authors answered all questions.

The revised manuscript meets the journal's requirements.

Author Response

Thank you for your guidance.

Round 2

Reviewer 2 Report

The reviewer thanks the authors for the much-improved manuscript; however, major issues have not been resolved sufficiently.

Major issue not resolved:

The authors have not sufficiently resolved the issue of repeat publication and have not justified how this current manuscript differs from the previous published work beyond the BAs section.

A similar project was done and is published by your group: Wang, D.B.; Zhao, L.H.; Su, R.N.; Jin, Y. Effects of different starter culture combinations on microbial counts and physico-chemical properties in dry fermented mutton sausages. Food Science Nutrition, 2019, 7, 1957-1968. 496.

The authors used the same strains, set-up, and came to the same conclusions except for the new data on BAs. This article even has the same write up layout, sample names, and even structuring of tables etc. The Authors did not discuss any results from this paper by Wang et al., 2019 and did not link it to any of their findings expect on similarity to methods and a statement “This result could be due to the decrease of aw [4,28]”. As recommended before, the authors need to restructure their article to highlight this previous work and discuss the current findings also referencing their previous work. Where similar results we obtained and where different results were observed needs to be highlighted. If this was a follow up study, please report it as such, focusing more on the new findings as it might seem as though the authors are trying to republish the same work twice. As previously stated, the new information on BAs is still very relevant.

Line 414: Table 4. Effect of mixed starter cultures on biogenic amines of dry fermented mutton sausage at fermentation and ripening stages.

The issue with Table 4, is not fully resolved. The authors need to account for the variability in day zero levels of TRY, PHE, PUT, CAD, HIS, and TYR either by calculating relative change factors or by applying a standardizing factor calculated based on the treatment with the highest day 0 value. That way the authors avoid over or underestimating the efficiency of the treatments they did in this study with the different starter cultures. Since this is the most critical part of their paper (novel data) the calculations and interpretations thereof are critical.

Please reevaluate or recalculate Table 4 data, factoring in day zero variability in your calculations.

Minor issues:

Line 91: please provide the references for these studies.

Line 103: since mutton and sheep tail fat came from the same batch of mutton sheep but not one single animal, the reviewers concern remains valid. These sheep although from the same batch will most likely have some differences in their microbiota. It is highly likely that using meat from different individual animals in part contributed to the day zero variations in BAs on Table 4 and MSA (Log CFU/g) derived numbers on Table 1 for CO (3.23±0.25Aa) and LB (5.25±0.02Ab) where no extra staphylococci were added.

Therefore, a statement along these lines is required: “Although mutton and sheep tail fat from the same batch of mutton sheep was used, some of the meat and fat was from different animals, consequently different bacterial species or numbers with different capacities to produce BAs might have been present, therefore, this data must be interpreted with this limitation in mind”.

Author Response

21nd October, 2021

Dear Circle He editor and Reviewers,

We would like to thank you and the Reviewers' for their constructive comments on our Manuscript foods-1417154. We have amended the manuscript based on the comments and highlighted our major or small changes in the text using red type and provided line numbers for those changes. Please see the revised manuscript and our point-by-point responses. We look forward to your further comments.

With best wishes,

Yours sincerely,

Debao Wang

Comments and Suggestions for Authors

The reviewer thanks the authors for the much-improved manuscript; however, major issues have not been resolved sufficiently.

Major issue not resolved:

The authors have not sufficiently resolved the issue of repeat publication and have not justified how this current manuscript differs from the previous published work beyond the BAs section.

A similar project was done and is published by your group: Wang, D.B.; Zhao, L.H.; Su, R.N.; Jin, Y. Effects of different starter culture combinations on microbial counts and physico-chemical properties in dry fermented mutton sausages. Food Science Nutrition, 2019, 7, 1957-1968. 496.

The authors used the same strains, set-up, and came to the same conclusions except for the new data on BAs. This article even has the same write up layout, sample names, and even structuring of tables etc. The Authors did not discuss any results from this paper by Wang et al., 2019 and did not link it to any of their findings expect on similarity to methods and a statement “This result could be due to the decrease of aw [4,28]”. As recommended before, the authors need to restructure their article to highlight this previous work and discuss the current findings also referencing their previous work. Where similar results we obtained and where different results were observed needs to be highlighted. If this was a follow up study, please report it as such, focusing more on the new findings as it might seem as though the authors are trying to republish the same work twice. As previously stated, the new information on BAs is still very relevant.

Response:Thank you for your guidance. We have made modifications and additions (lines 261-263, 284-291, 354-357, 383-384, 392-394, 407-410, and 419-421).

Line 414: Table 4. Effect of mixed starter cultures on biogenic amines of dry fermented mutton sausage at fermentation and ripening stages.

The issue with Table 4, is not fully resolved. The authors need to account for the variability in day zero levels of TRY, PHE, PUT, CAD, HIS, and TYR either by calculating relative change factors or by applying a standardizing factor calculated based on the treatment with the highest day 0 value. That way the authors avoid over or underestimating the efficiency of the treatments they did in this study with the different starter cultures. Since this is the most critical part of their paper (novel data) the calculations and interpretations thereof are critical.

Please reevaluate or recalculate Table 4 data, factoring in day zero variability in your calculations.

Response:Thank you for your suggestion. We reevaluated the data in Table 4 by calculating the changes in all groups of data relative to day 0, as 3.5. BAs.

Minor issues:

Line 91: please provide the references for these studies.

Response:Thank you for your guidance. We have added the related reference.

Line 103: since mutton and sheep tail fat came from the same batch of mutton sheep but not one single animal, the reviewers concern remains valid. These sheep although from the same batch will most likely have some differences in their microbiota. It is highly likely that using meat from different individual animals in part contributed to the day zero variations in BAs on Table 4 and MSA (Log CFU/g) derived numbers on Table 1 for CO (3.23±0.25Aa) and LB (5.25±0.02Ab) where no extra staphylococci were added.

Therefore, a statement along these lines is required: “Although mutton and sheep tail fat from the same batch of mutton sheep was used, some of the meat and fat was from different animals, consequently different bacterial species or numbers with different capacities to produce BAs might have been present, therefore, this data must be interpreted with this limitation in mind”.

Response:This statement has been added in lines 100-102.

This manuscript is a resubmission of an earlier submission. The following is a list of the peer review reports and author responses from that submission.

Round 1

Reviewer 1 Report

I did a bibliographic research and I think that your manuscript is very similar to another already published by your research group (Wang et al - Effects of different starter culture combinations on microbial counts and physico-chemical properties in dry fermented mutton sausages)

 The experiments performed in this work are almost the same of another work, so for me it lacks of novelty .
In details:

    they used the same combination of bacterial strains in the analized samples (CO batch, no starter cultures, used as control; LB batch with Lactobacillus sakei; LS batch with L. sakei + Staphylococcus xylosus; and LSS batch with L. sakei + S. xylosus + Staphylococcus carnosus) and also the same matrix (mutton sausage) 

    they made in both these same analyses: pH, water activity, thiobarbituric acidreactive substances (TBARS) analysis, Free amino acid analysis
    the only "novelty" is the analysis of biogenic amines that, in my opinion, is not sufficient to justify a new publication 

Reviewer 2 Report

The manuscript presented by Wang et. al. exposes the study of several microorganisms as fermentation starters in the manufacture of sausages, attending to the modifications caused in the physicochemical profile and levels of free amino acids and biogenic amines. The subject that the manuscript deals with, although not very innovative, is interesting and may be a relevant contribution to the field of the reduction of biogenic amines in matured foods.

While the manuscript is essentially fine, and scientifically correct, a thorough review of the English language and style is mandatory. In fact, the errors are too numerous to list here, but there are conflicts in the use of verb tenses, formal academic writing, loss of meaning of sentences due to inappropriate writing, repetitions that make reading very heavy, use of different font sizes… For this reason, it is absolutely necessary to review the document by a native English speaker, preferably by a professional editing service.

Authors should always include the full name the first time initials are used (eg in lines 65, 70 and 79) (this applies to both the manuscript and the abstract, which must be stand-alone).

Line 56: The FDA reference is very old, isn't there a more modern bibliography on it?

Line 60: point (1) is not clear

Line 74: Add examples of these studies (cite).

Line 91: What do you mean by identification by starter safety?

Lines 98-100: What medium did you use for the growth of each strain? As written, it appears that S. xylosus was grown on MRS, which seems unlikely. The wording of the text needs to be vastly improved. Number of hours of cultivation, temperature, agitation?

Line 113: 1000 g. of what?

Lines 118-121: MRS is not a selective medium for lactic acid bacteria, since although it favors their growth, it cannot prevent other microorganisms from growing (which can easily be verified by inoculating MRS with non-lactic bacteria in the laboratory), therefore LAB counts are invalid.

Line 136: please replace rmp with rpm

Line 159: What is r/min?

Table footer 1: you talk about the superscripts a, b, c and d, however, d does not appear in the table anywhere.

Lines 188-189: the number of LAB is not close to that of Enterobacteriaceae

Line 193: it cannot be stated that the decrease in LAB (although it has already been indicated that the LAB counts are not valid) is due to a decrease in pH, when precisely the decrease would be due to the production of acids by the LAB themselves. In addition, later it is stated that from day 5 to day 8 the pH rises in LS and LSS, contradicting this. It seems as if different people had participated in the writing of the manuscript, without looking at what was written by other co-authors.

Lines 202-203: this does not make sense. Please revise.

Figure 1: In the legend, LSS is repeated and LS is missing

Conclusions: they are a mere repetition of results and not true conclusions, so they should be rewritten.

Reviewer 3 Report

Comments and Suggestions for Authors

General comments

The authors compared the effects of different starter cultures as single cultures or mixed starters on the commensal microbiota, inhibition of biogenic amine formation, and product characteristics in dry fermented mutton sausage. They showed that mixed starter cultures have antimicrobial activity against commensal microbiota and reduced the production of BAs.  Suggesting that mixed starter cultures have potential application in fermented meat products, improving their safety.

Strengths:

BAs in sausages represent a health risk for consumers, thus investigating the BAs accumulation mechanisms and ways to inhibit them is important to improve fermented sausage safety. The study is thought-provoking and adds more understanding and options to limit BA accumulation in fermented sausages. The study also helps in the identification and development of candidates for BAs control in fermented meat products. The use of several approaches to try and establish the benefits of mixed starter cultures is a great plus for the paper.

Suggestions to the authors:

The paper needs language editing as some points are not coming out clearly, which detracts from the overall good work done. The abstract needs to be restructured to emphasize the important findings and to highlight why the study was done. There are several papers that detail work on this research topic, results obtained in this present study need to be discussed with some reference to these studies. A few examples are provided below (References).

The authors also need to highlight the effects of changing starter cultures might have on product characteristics such as flavor. The introduction doesn’t fully introduce the topic, for instance, the negative health effects of BAs are not highlighted. These need to be discussed. Also, the effects of moderate BAs concentration in people with underlying conditions or who are taking certain medications need to be highlighted. Under normal conditions, the human body can detoxify BAs ingested from foods by acetylation and oxidation by monoamine oxidase (MAO) and diamine oxidase (DAO), and specific amine methyltransferases. However, this detoxification system can be negatively influenced by some food components (other amines, alcohol and its metabolite acetaldehyde, phenols, etc.), drugs acting as inhibitors of MAO and DAO, and tobacco.

My main concerns are about how materials and methods are described and, consequently, about how results are displayed, interpreted, and discussed. Some details are missing or conflicting and should be specified. For instance, in PI analysis, extraction was repeated 1–2 times, but the sample on which extraction was done once or twice is not specified. It’s not clear if meat or fat from the same animal was used in the preparation of all the sausages. If it was not from the same animal, it might have different commensal microbiota with differing capabilities of BAs production. This needs to be clearly defined. If meat or fat from different animals was used, a statement that such data must be interpreted with caution due to this weakness needs to be included.

Data analysis: Because the starting amounts of BAs or CFU counts of commensal microbiota were different in some samples, the authors should calculate the growth potential observed for each treatment or relative changes to each value for the analysis to account for the initial variation of each sample (e.g., “day 0 - day 8 CFU counts” or “day 8 BA/day0 BA”). For instance, in Table 4, PHE of CO on day 0 was 1.25±0.04 compared to 0.45±0.06 on LB, 0.44±0.06 on LS, and 0.84±0.17 on LSS. Relative change factors are CO (2.904), LB (7.2), LS (6.205), and LSS (3.238). Meaning that relative to starting value of PHE on day 0, CO had the least increase in this BA.

Results and discussion. The results need to be discussed in more detail and the possible reasons for the differences need to be explored more. It would be prudent for the authors to highlight their study setup limitations.

References

  1. Latorre-Moratalla ML, Bover-Cid S, Veciana-Nogués MT and Vidal-Carou MC. 2012. Control of biogenic amines in fermented sausages: role of starter cultures. Front. Microbio. 3:169. doi: 10.3389/fmicb.2012.00169.
  2. Li L, Zou D, Ruan L, Wen Z, Chen S, Xu L and Wei X. 2019. Evaluation of the Biogenic Amines and Microbial Contribution in Traditional Chinese Sausages. Front. Microbiol. 10:872. doi: 10.3389/fmicb.2019.00872.
  3. Pasini F, Soglia F, Petracci M, Caboni MF, Marziali S, Montanari C, Gardini F, Grazia L, Tabanelli G. 2018. Effect of Fermentation with Different Lactic Acid Bacteria Starter Cultures on Biogenic Amine Content and Ripening Patterns in Dry Fermented Sausages. Nutrients. 10(10):1497. https://doi.org/10.3390/nu10101497.
  4. Chong Xie, Hu-Hu Wang, Xiao-Kai Nie, Lin Chen, Shao-Lin Deng & Xing-Lian Xu. 2015. Reduction of biogenic amine concentration in fermented sausage by selected starter cultures, CyTA - Journal of Food, 13:4, 491-497, DOI: 10.1080/19476337.2015.1005027
  5. Jirasak Kongkiattikajorn. 2015. Potential of starter culture to reduce biogenic amines accumulation in som-fug, a Thai traditional fermented fish sausage. https://doi.org/10.1016/j.jef.2015.11.005
  6. Kim Hyeong Sang, Lee Seung Yun, Kang Hea Jin, Joo Seon-Tea, Hur Sun Jin. 2019. Effects of Six Different Starter Cultures on Mutagenicity and Biogenic Amine Concentrations in Fermented Sausages Treated with Vitamins C and E. Food Sci Anim Resour. 39(6):877-887. https://doi.org/10.5851/kosfa.2019.e66.
  7. Bover-Cid, S., Hugas, M., Izquierdo-Pulido, M., and Vidal-Carou, M. C. 2000. Reduction of biogenic amine formation using a negative amino acid-decarboxylase starter culture for fermentation of fuet sausages. J. Food Prot. 63, 237–243.
  8. Bover-Cid, S., Izquierdo-Pulido, M., and Vidal-Carou, M. C. 1999. Effect of proteolytic starter cultures of Staphylococcus on biogenic amine formation during the ripening of dry fermented sausages. Int. J. Food Microbiol. 46, 95–104.

Specific comments

Abstract

Line 21: Please define BAs write it as “biogenic amines (BAs)”.

Line 22-23: According to Table 1, LSS addition only increased staphylococci numbers, for lactic acid bacteria (LAB) it only significantly increased them until day 2. On day 8 LSS had the lowest LAB numbers. The statement “Inoculation of the mixed starter cultures increased the number of lactic acid bacteria and staphylococci in sausages” is therefore not in agreement with your results. Please rephrase it.

Line 24: Please replace “water evaporation” with “water activity reduction”.

Line 25-26: Please rephrase to “Compared with the CO, the mixed starter effectively inhibited Enterobacteriaceae. At the end of ripening the LSS group had about 1.25 CFU/g less than the CO group.

Line 26: Is it 1.77 CFU/g or 1.74 CFU/g?

Line 29-33: Consider rephrasing as it is not clear. Suggestion: “The level of histamine, cadaverine, and putrescine, common BAs showed an opposite trend to the increase of the corresponding precursor amino acid content, which were significantly lower (P < 0.001) in the LS and LSS sausages than in CO. Overall this study shows that L. sakei 3X-2B + S. xylosus SZ-8 + S. carnosus SZ-2 is a potential mixed starter for application in fermented meat products.”

Introduction

The introduction needs language editing as some points are not coming out clearly which detracts from the overall great work done in generating it. Please also mention other BAs reduction strategies that are being applied in fermented meat production.

Line 37-38: Please provide a supporting reference.

Line 38-39: please rephrase to make it clear.

Line 40-41: Consider replacing “decomposed” with “broken down” or “metabolized”. Please consider this throughout the paper.

Line 46-49: The effects of ingestion of moderate BAs concentration in people with underlying conditions or who are taking certain medications needs to be highlighted. I agree, under normal conditions, the human body can detoxify BAs ingested from foods by acetylation and oxidation by monoamine oxidase (MAO) and diamine oxidase (DAO), and specific amine methyltransferases. However, this detoxification system can be negatively influenced by some food components (other amines, alcohol and its metabolite acetaldehyde, phenols, etc.), drugs acting as inhibitors of MAO and DAO, and tobacco.

Line 49-51: Please rephrase to make it clear. It seems the authors want to highlight the potential interaction or reaction of BAs and nitrites to form nitrosamine, but this is not clear with the way this statement has been structured.

Line 56: (FDA, 1990); Please use the same referencing system and add this reference to the reference list.

Line 60: Please rephrase to “are often taken: (1) Reducing the level of the biogenic amines precursors [12],”

Line 65: Please consider adding this reference: Latorre-Moratalla et al., 2012. Control of biogenic amines in fermented sausages: role of starter cultures. Front. Microbio. 3:169. doi: 10.3389/fmicb.2012.00169.

Line 76: Please rephrase to “This work was done to evaluate the effect of different mixed starter cultures”

Methods

Line 91-92: Please rephrase to “Three isolates were identified by starter safety and the sequencing of 16S ribosomal RNA gene.”

How was the starter safety determined especially if whole-genome sequencing was not done? For practical application on products to be consumed, is 16S rRNA sequencing only sufficient to determine safety for such starter cultures?

Line 94: It is not clear if meat or fat from the same animal was used in the preparation of all the sausages. If it was not from the same animal, it might have different commensal microbiota with differing capabilities of BAs production. This needs to be clearly defined. If meat or fat from different animals was used then a statement that such data must be interpreted with caution due to this weakness needs to be included.

Line 98: It is not clear if solid agar or liquid media was used. “using MRS (De Man Rogosa Sharpe) and MSA (Mannitol salt agar) media, respectively”. If agar was used how was 108 CFU/mL achieved?

Line 99 & 102: How were these 108 CFU/mL or 107 CFU/g confirmed?

Line 110: what was the average weight of each sausage?

Line 107-114: This section requires language editing.

Line 156: Were the MRS Agar plates incubated anaerobically or aerobically and what was the duration of incubation? Would incubation anaerobically at say 30 oC for 72 hours alter the bacteria that would have grown on these plates?

Line 138: For PI analysis, extraction was repeated 1–2 times. However, the sample on which extraction was done once or twice is not specified. I think comparing a sample that was extracted once to samples with double extraction would introduce bias.

Line 147, 150, 152: What was the actual weight of the starting sausage sample used and the final weight achieved after drying? “A certain amount”, “More than 1 mL of hydrolysate was blown dry by nitrogen”, please provide the actual sample volumes used for these stages of the protocol.

Line 160: what was used to filter the supernatant?

Line 162: replace “trichloroacetic acid” with “TCA”.

Line 178: Please change from “3. Results” to “3. Results and Discussion”.

The results need to be discussed in more detail referencing other similar studies which might offer supporting information to the reported findings. See example references provided above.

Line 180: According to Table 1, LSS addition only increased staphylococci numbers, for lactic acid bacteria (LAB) it only significantly increased them until day 2. On day 8 LSS had the lowest LAB numbers. The statement “Inoculation of the mixed starter cultures increased the number of lactic acid bacteria and staphylococci in sausages” in the abstract is therefore not in agreement with your results. Please rephrase it.

The growth potential of LAB was greater on CO compared to LB, LS, and LSS, probably due to the absence of competition from the started cultures.

Please comment on the “Jamerson effect” in relation to your observations.

Line 188-189: What do the authors mean by “all groups increased first and then decreased and were close to the number of Enterobacteriaceae”?

Line 193: would the pH decrease affect the LAB? Where is this data derived from (r=0.66, P < 0.05)?

Line 198-201: Please rephrase to “Staphylococci contributes to proteolytic and lipolytic responses, which are of great significance in improving the color, texture, and flavor, as well as inhibiting lipid peroxidation of fermented meat products [27–28].”

Line 211 & 267: Fig 1 A and B and Fig 2 A and B, please correct the figure legends LS is missing and LSS appears twice.

Line 216. LS and LSS had lower water activity on days 5 and 8. How do starter cultures alter water activity? Please discuss how mixed starter cultures contribute to water activity reduction.

Line 227: Table 2: please replace the letters in the colour column with the actual words “lightness (L), redness (a), yellowness (b)” as these can lead to confusion with the letters used for statistical significance. 

What could be the reason for the significant difference in lightness of LB on day 5?

What could be the reason for the significant difference in yellowness of LSS on days 0 and 2?

Line 238-242: Was the reduction of Micrococcaceae counts established in this study? If it was not determined, please rephrase this section accordingly.

Line 308-310: Please rephrase to make it clearer.

Line 313-316: Please rephrase (end of pickling, P < 0.001) to (end of fermentation, P < 0.001).

Line 318: Please rephrase to “was considered detrimental to human health [49].”

Line 342-344: How accurate is this statement if you consider the relative changes to PHE? Also, do the mixed starters inhibit the enzyme or the bacteria producing the enzyme?

Table 4, PHE of CO on day 0 was 1.25±0.04 compared to 0.45±0.06 on LB, 0.44±0.06 on LS, and 0.84±0.17 on LSS. Relative change factors are CO (2.904), LB (7.2), LS (6.205), and LSS (3.238). Meaning that relative to starting value of PHE on day 0, CO had the least increase in this BA. A similar effect might be observed on TRY and TYR.

Please calculate the relative changes for all the other BAs and them make the appropriate comment on inhibition of BAs production.

Reviewer 4 Report

The paper entitled " Effect of mixed starters on proteolysis and formation of biogenic amine in dry fermented mutton sausage " describes the effect of the starter cultures on selected parameters of the dry mutton sausages. The topic of the paper is interesting and important. The described research, as well as used methods, justify a choice of the journal. The introduction provides sufficient background. The work is properly designed and performed. However, I have some questions and suggestions.

  1. The main factor determining the formation of BAs in food is the presence of free amino acids, proteins and appropriate microorganisms. Did the authors consider the possibility that the content of free amino acids in LS and LSS samples could cause a significant increase of BAs content in the process of storing sausages?
  2. Did the authors analyze the discussed parameters (FAAS, BAs, pH etc) and microorganisms in the starting material (lean meat on hind legs and tail fat)?
  3. Table 1. Lack of units. Row: “Sign”. is not readable. There is no extensive discussion of the amount of MSA and VRBGA s in the LS and LSS batch (day 5 and 8).
  4. According to Figure 1A, a large reduction in pH on the second day for LSS is observed. How can this be explained?
  5. More information about quantitative analysis of FAAs and BAs should be added
  6. Line 79 abbreviation “TVBN” should be explained